

# Pollination implications of the diverse diet of tropical nectar-feeding bats roosting in an urban cave

Voon-Ching Lim[1,2,*], Rosli Ramli[2], Subha Bhassu[2,3] and John-James Wilson[4,5,*]

[1] Rimba, Kuala Lumpur, Malaysia
[2] Institute of Biological Sciences, Faculty of Science, University of Malaya, Kuala Lumpur, Malaysia
[3] Centre for Research in Biotechnology for Agriculture, University of Malaya, Kuala Lumpur, Malaysia
[4] School of Applied Sciences, Faculty of Computing, Engineering and Science, University of South Wales, Pontypridd, UK
[5] Department of Microbiology and Parasitology, Faculty of Medical Science, Naresuan University, Phitsanulok, Thailand
* These authors contributed equally to this work.

Corresponding author
Voon-Ching Lim,
voonchinglim@siswa.um.edu.my,
voonchinglim@gmail.com

## ABSTRACT

**Background:** Intense landscaping often alters the plant composition in urban areas. Knowing which plant species that pollinators are visiting in urban areas is necessary for understanding how landscaping impacts biodiversity and associated ecosystem services. The cave nectar bat, *Eonycteris spelaea*, is an important pollinator for many plants and is often recorded in human-dominated habitats. Previous studies of the diet of *E. spelaea* relied on morphological identification of pollen grains found in faeces and on the body of bats and by necessity disregarded other forms of digested plant material present in the faeces (i.e., plant juice and remnants). The main objective of this study was to examine the diet of the nectarivorous bat, *E. spelaea,* roosting in an urban cave at Batu Caves, Peninsular Malaysia by identifying the plant material present in the faeces of bats using DNA metabarcoding.

**Methods:** Faeces were collected under the roost of *E. spelaea* once a week from December 2015 to March 2016. Plant DNA was extracted from the faeces, Polymerase chain reaction (PCR) amplified at *ITS2* and *rbcL* regions and mass sequenced. The resultant plant operational taxonomic units were searched against NCBI GenBank for identification.

**Results:** A total of 55 species of plants were detected from faeces of *E. spelaea* including *Artocarpus heterophyllus, Duabanga grandiflora* and *Musa* spp. which are likely to be important food resources for the cave nectar bat.

**Discussion:** Many native plant species that had not been reported in previous dietary studies of *E. spelaea* were detected in this study including *Bauhinia strychnoidea* and *Urophyllum leucophlaeum,* suggesting that *E. spelaea* remains a crucial pollinator for these plants even in highly disturbed habitats. The detection of many introduced plant species in the bat faeces indicates that *E. spelaea* are exploiting them, particularly *Xanthostemon chrysanthus,* as food resources in urban area. Commercial food crops were detected from all of the faecal samples, suggesting

that *E. spelaea* feed predominantly on the crops particularly jackfruit and banana and play a significant role in pollination of economically important plants. Ferns and figs were also detected in the faeces of *E. spelaea* suggesting future research avenues to determine whether the 'specialised nectarivorous' *E. spelaea* feed opportunistically on other parts of plants.

## INTRODUCTION

Urban land cover in Peninsular Malaysia expanded from 4,644.3 km$^2$ in year 2000 to 5,364.4 km$^2$ in year 2010 with an average annual increase of 1.5% (*Schneider et al., 2015*). Intense landscaping often increases the species richness and homogeneity of plants in these urban areas (*Grimm et al., 2008*; *Kowarik, 2011*). These plants support diverse assemblages of birds and bats (*Corlett, 2005*; *Aida et al., 2016*), which in turn provide seed dispersal and pollination services, and consequently aid in maintaining urban green spaces (*Tan, Zubaid & Kunz, 2000*; *Corlett, 2005*; *Sheherazade, Pradana & Tsang, 2017*). However, the preference for planting non-native species in parks and household gardens for urban beautification and food may create competition for pollination services which could affect the reproductive success and survival of native plants (*Faeth et al., 2005*). Knowing which plants pollinators are visiting in urban areas is essential for assessing how planting schemes will affect biodiversity and associated ecosystem services.

The cave nectar bat, *Eonycteris spelaea* (family: Pteropodidae), is generally categorised as specialised nectarivorous bat (*Fleming, Geiselman & Kress, 2009*; *Stewart & Dudash, 2017*) that feeds on nectar and pollen, and consequently provides pollination services (*Srithongchuay, Bumrungsri & Sripao-raya, 2008*; *Bumrungsri et al., 2009*; *Acharya et al., 2015a*; *Nor Zalipah et al., 2016*). *E. spelaea* is one of three nectarivorous bats present in Peninsular Malaysia and is often recorded in urban and agricultural areas (*Lim et al., 2017*). The capability of *E. spelaea* to travel long distances for food and visit night-blooming plants with high frequency likely contributes to an important role as a pollinator (*Start & Marshall, 1976*; *Stewart & Dudash, 2017*).

The diet of *E. spelaea* in Southeast Asia was previously assessed through morphological identification of pollen grains (found in faeces and on the body of bats) examined microscopically. *Start & Marshall (1976)* observed 31 distinct types of pollen in faeces of *E. spelaea* collected under two roosts at Batu Caves and Gua Sanding in Peninsular Malaysia but could only identify the pollen grains of 17 plant species. *Bumrungsri et al. (2013)* collected 11 types of pollen from captured individuals of *E. spelaea* at Khao Kao Cave in Thailand but could only identify the pollen grains of four plant species. Similarly, *Thavry et al. (2017)* recorded 13 types of pollen in faeces of a roosting colony at Bat Khteas Cave in Cambodia but could only identify the pollen grains of four plant species. The lack of distinctive morphological characters on pollen grains (*Bell et al., 2016*) and lack of reference specimens for comparison (*Aziz et al., 2017a*) likely account for

the difficulties in identification of plant species. Furthermore, these studies prioritised pollen grains (solid plant material which are physically identifiable in faeces) and by necessity disregarded other types of plant material defecated by the bats (i.e., nectar and leaf fragments). As *E. spelaea* feeds mainly on nectar and pollen (*Fleming, Geiselman & Kress, 2009*; *Stewart & Dudash, 2017*), and possibly on fruits and leaves (*Start & Marshall, 1976*; *Bumrungsri et al., 2013*), it is necessary to identify all the plant material present in the faeces in order to have a complete picture of the ecological role of the cave nectar bat.

DNA barcoding (*Hebert, Cywinska & Ball, 2003*) can aid in identification of digested plant material in faeces of bats (*Hayward, 2013*) without requiring the high level of taxonomic expertise necessary for microscopic identification of pollen grains (*Pompanon et al., 2012*). Plant DNA can be extracted from faeces, Polymerase chain reaction (PCR) amplified with taxon-specific universal PCR primers (e.g., *rbcL* and *ITS2* (*CBOL Plant Working Group, 2009*; *Chen et al., 2010*)), and sequenced to recover short DNA sequences which can be matched to taxonomically verified sequences for species identification (*Pompanon et al., 2012*). Recent advances in high-throughput sequencing platforms have introduced DNA metabarcoding which involves simultaneous DNA sequencing of multiple templates in complex samples (e.g., faeces) and allows detection of multiple species in a single sample (*Pompanon et al., 2012*; *Brandon-Mong et al., 2015*). DNA metabarcoding has been widely used to identify the diet of honey bees (*de Vere et al., 2017*), omnivorous brown bears (*De Barba et al., 2014*), large herbivores (*Kartzinel et al., 2015*) and insectivorous (*Clare et al., 2014*) and frugi-nectarivorous bats (*Aziz et al., 2017a*).

The aim of this study was to examine the diet of *E. spelaea* roosting in an urban cave in Peninsular Malaysia by using DNA metabarcoding to identify the digested plant material in bat faeces. Specifically, we asked whether *E. spelaea* (i) feeds primarily on native plants and still serves as their crucial pollinator in an urban area, or (ii) exploits introduced plant species (which are commonly planted for food and urban beautification) as food resources, potentially pollinating them and impacting the reproductive success of native plants.

## MATERIALS AND METHODS

### Ethics

Faecal collection was conducted at Dark Cave, Batu Caves with authorization from the Department of Wildlife and National Parks, Peninsular Malaysia (Ref: JPHL&TN(IP) 100-34/1.24 Jld. 4(34)), the Malaysian Nature Society and Majlis Perbandaran Selayang (Ref: Bil(35)dlm.MPS 3/3-117/153 JL) using a protocol approved by the Institutional Animal Care and Use Committee, University of Malaya (Ref: ISB/10/06/2016/LVC (R)).

### Study site and bat species

Batu Caves constitute an extensive karst cave system developed within an isolated 329 m high limestone massif located in Gombak District, part of the Klang Valley conurbation in Selangor state adjacent to Kuala Lumpur Federal Territory (*Moseley, Lim & Lim, 2012*; *Grismer et al., 2014*). Batu Caves is surrounded by industrial parks and

residential areas (*Grismer et al., 2014*) and includes a Hindu temple that has become a major tourist attraction (*Kasim, 2011*). The cave complex includes the Dark Cave, a protected cavern with >2,000 m of passages (*Price, 2002*) managed by the Cave Management Group under the Malaysian Nature Society (http://www.darkcavemalaysia.com/). Dark Cave is an ecologically diverse karst cave system which supports a large number of animals (*Moseley, 2009*; *Moseley, Lim & Lim, 2012*) including a colony of *E. spelaea*. *Start & Marshall (1976)* estimated that the colony comprised >10,000 individuals whereas *Beck & Lim (1972)* and *Gould (1988)* estimated >4,000 individuals. For this study, faecal samples were collected under the *E. spelaea* roost at Dark Cave (Fig. 1).

## Faecal collection

Approximately 10 ml of fresh faecal samples were collected non-invasively under the roost of *E. spelaea* once every week from 31 December 2015 to 4 March 2016 (i.e., 10 days over 10 weeks). Overall, a total of ~100 ml of fresh faecal material was collected and used for the study. As the Cave Management Group cleans the floor below the roost daily to prevent the accumulation of bat faeces (which is unappealing to tourists), faeces below the roost were assumed to be deposited the previous night. The faeces were kept in 1.5 ml tubes filled with 99.8% ethanol and stored at −20 °C prior to analysis (following *Lim et al., 2018*).

## Preparation of faecal samples

The faeces were centrifuged to form pellets and the supernatant were discarded. The pellets were incubated at 56 °C for 2 h to evaporate moisture (i.e., ethanol), pooled according to collection week and homogenised using a TissueLyser II (Qiagen, Hilden, Germany) with 3 mm tungsten carbide beads (Qiagen, Hilden, Germany) for 4 min at 30 1/s.

## Plant DNA extraction, PCR amplification, clean-up and sequencing

DNA extraction was performed twice using the QIAamp DNA stool mini kit (Qiagen, Hilden, Germany) following the manufacturer's protocol which resulted in two DNA replicates for each weekly sample. The purity and concentration of the DNA was examined with NanoDrop 2000c UV–Vis Spectrophotometer (Thermo Fisher Scientific, Waltham, MA, USA). DNA extracts with a purity range from 1.8 to 1.9 and concentrations ≥50 ng/μl were used for PCR amplification.

Two DNA barcode markers were selected for this study: *rbcL* due to its relative universality (i.e., universal primers; *CBOL Plant Working Group, 2009*) and *ITS2* due its higher taxonomic resolution (*Chen et al., 2010*). Both markers have a large number of taxonomically verified DNA reference sequences available in NCBI GenBank (http://www.ncbi.nlm.nih.gov/) (*rbcL* = 155,634; *ITS2* = 243,155; *Bell et al., 2016*) and have been used successfully to examine the diet of rolled-leaf beetles in a tropical rainforest in Costa Rica (*García-Robledo et al., 2013*) and the plant sources of honey (*Prosser & Hebert, 2017*).

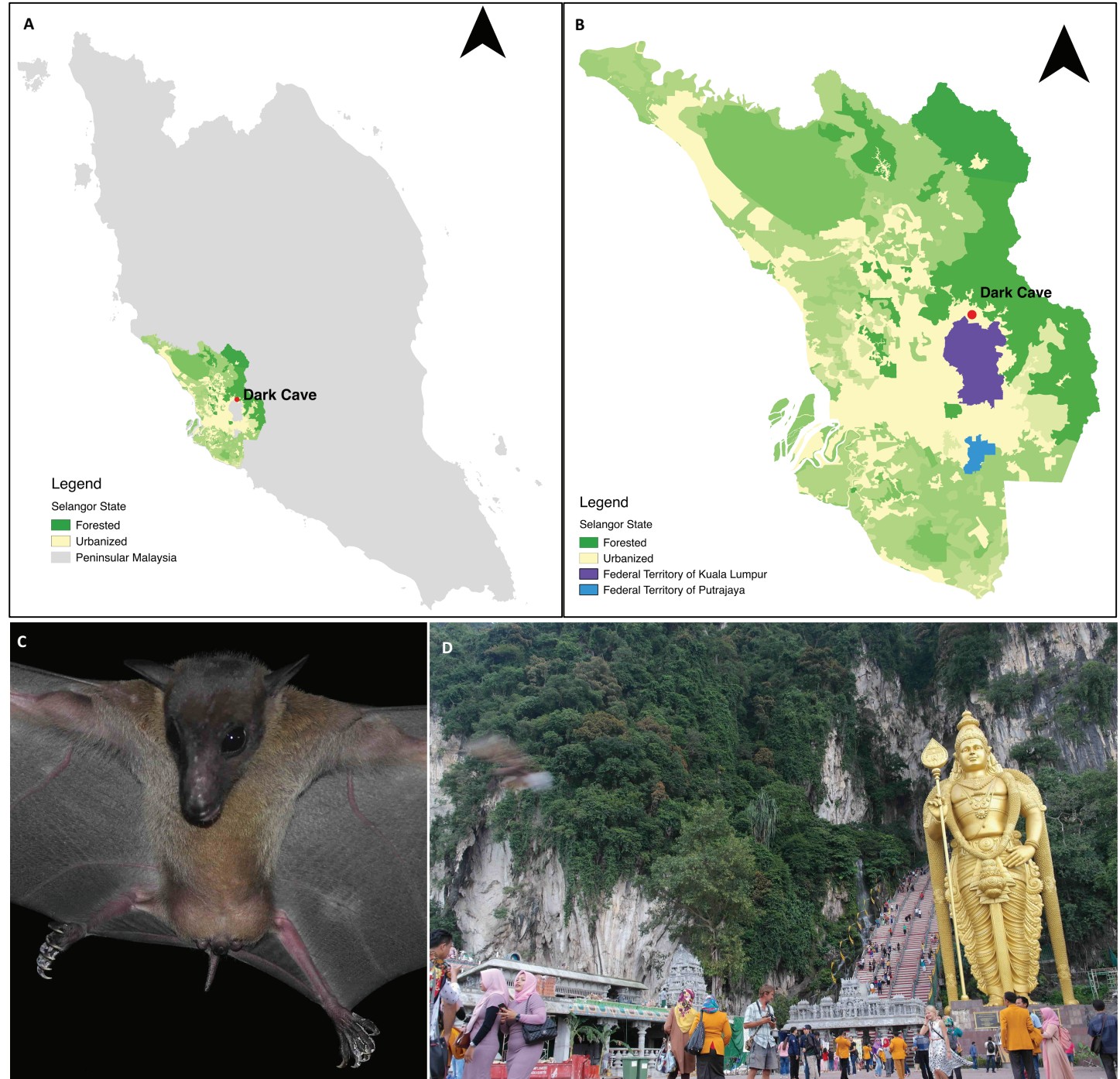

**Figure 1** **A permanent roosting colony of *Eonycteris spelaea* was located at Dark Cave Conservation Site, one of the caves in Batu Caves.** (A) The location of Dark Cave Conservation Site in Peninsular Malaysia. (B) Land cover of Selangor state where Dark Cave is located (source: http://www.globalforestwatch.org/). (C) Close-up of *E. spelaea* taken by VCL. (D) Batu Caves serves as temple for Hindu prayers and tourist attraction for its cultural and natural heritage, photographed by VCL.

Fragments of *rbcL* and *ITS2* were amplified using universal primers with Illumina adaptors (Supplemental Information 1). Five PCR amplifications were performed for each DNA extract replicate together with one positive (*Musa* sp.) and one negative (ddH$_2$O) control. Each PCR amplification was performed in a total volume of 25 μl consisting of 12.5 μl EconoTaq PLUS GREEN 2X Master Mix (Lucigen, Middleton, WI, USA), 0.25 μl of each forward and reverse primer (100 μM), 7–8 μl of ddH$_2$O and 4–5 μl of DNA. The thermocycling profile for *rbcL* was: initial denaturation at 95 °C for 2 min, denaturation, annealing and extension at 95 °C for 30 s, 55 °C for 30 s, 72 °C for 10 s (35 cycles) and a final extension at 72 °C for 6 min. The thermocycling profile for *ITS2* was: initial denaturation at 94 °C for 2 min, denaturation, annealing and extension of 94 °C at 30 s, 55 °C at 30 s, 72 °C at 20 s (35 cycles) and a final extension at 72 °C for 10 min.

PCR products were checked on 2% agarose gels and extracted and purified using a NucleoSpin Gel and PCR Clean-up kit (Macherey-Nagel, Düren, Germany) following the manufacturer's instructions. The purified products were assessed with a NanoDrop 2000c UV–Vis Spectrophotometer (Thermo Fisher Scientific, Waltham, MA, USA). Products with purity ranging from 1.8 to 1.9 and concentration ≥50 ng/μl were used for a second round of PCR to generate amplicons containing dual-index multiplex identifier (MID) tags and sequencing on an Illumina Miseq Sequencer (Illumina, San Diego, CA, USA) with 2 × 300 bp paired-end read setting.

Paired-end reads were sorted into datasets (i.e., weeks) by MID and merged (for *ITS2*). *RbcL* reads could not be merged due to the lack of overlapping sequence between paired-end reads. Therefore, *rbcL* reads containing only the forward primer were used in subsequent steps as these sets of reads were longer and more abundant.

### Filtering pipeline

Using the Galaxy web server (https://usegalaxy.org/; *Giardine et al., 2005*), files were converted to Illumina1.8+ format using 'FASTQ Groomer' (*Blankenberg et al., 2010*). Primers were removed using 'Clip' (http://hannonlab.cshl.edu/fastx_toolkit/). Short (*rbcL* < 100 bp; *ITS2* < 320 bp) and low quality (QV < 20) reads were discarded using 'Filter FASTQ' (*Blankenberg et al., 2010*). Remaining reads were de-replicated with 100% identity using 'VSearch dereplication' (*Rognes, Mahé & Xflouris, 2015*). Duplicates and possible chimeras were then removed using 'cd-hit-dup' (*Fu et al., 2012*). Remaining reads were clustered into operational taxonomic units (OTU) with 98% identity using 'VSearch clustering' (*Rognes, Mahé & Xflouris, 2015*).

### Assigning taxonomic names

Operational taxonomic units were BLAST-ed against NCBI GenBank (https://blast.ncbi.nlm.nih.gov/Blast.cgi; *Boratyn et al., 2013*) with the following Megablast parameters: identity = 100%, minimum score = 300 and maximum expected value = 0.01. Taxonomic names were assigned to OTU using the following criteria: (i) when the OTU matched to records from one species only, the species name was assigned; (ii) when the OTU matched to records from multiple species from one genus only, the

genus name was assigned; (iii) when the OTU matched to records from multiple genera belonging to one family only, the family name was assigned.

Taxonomic names were checked against *Corner (1997)* and *Boo, Chew & Yong (2014)* for the local uses of the species (e.g., food, medicinal and aesthetic), and against the Catalogue of Life (http://www.catalogueoflife.org) for the status of the species as native or introduced. To assess whether the species potentially provide stable (i.e., flower throughout the year) or alternative (i.e., flower seasonally) food resources to the cave nectar bats, the taxonomic names were checked against local literature for information regarding the flowering phenology: *Flora of the Malay Peninsula* (*Ridley, 1922–1925*), *Tree flora of Malaya: a manual for foresters* (*Whitmore, 1972–1989*), *Wayside trees of Malaya* (*Corner, 1997*), *Flora of Peninsular Malaysia* (*Kiew et al., 2010–2015*; *Parris et al., 2010–2013*) and *Plants in Tropical Cities* (*Boo, Chew & Yong, 2014*). See Supplemental Information 2 for further details on the flowering phenology of each taxonomic names.

## Species richness and sampling completeness ratio

All analyses were performed using R version 3.3.1. (*R Core Team, 2017*). The detection of plant species in faecal samples of *E. spelaea* was recorded as absent or present following *Prosser & Hebert (2017)*. Currently, DNA metabarcoding cannot be considered quantitative due to biological (e.g., varying copy numbers of plastid and nuclear DNA in pollen among and within species; *Bell et al., 2016*) and methodological (e.g., PCR amplification bias caused by universal primers; *Prosser & Hebert, 2017*) factors. The species richness and the sampling completeness ratio were estimated using the SpadeR package (*Chao & Shen, 2010*). Chao2 is more suitable for the incidence-type data collected in this study as it estimates the species richness based on the incidence of each species (i.e., presence or absence) recorded in each sampling unit (*Chao & Chiu, 2016*). Several Chao2 models were used to assess consistency of estimates provided by each model. A homogeneous model was also included under the assumption that all plant species have the same detection probabilities, but usually severely underestimates the true species richness if heterogeneity exists (*Chao & Chiu, 2016*). Rarefaction and extrapolation sampling curves of estimated species richness and the sampling completeness ratio were created using the iNEXT package (*Hsieh, Ma & Chao, 2016*) with Chao2 and a 95% confidence interval (R scripts are available as Supplemental Information 3).

## Relative detection rate of each plant species in faeces of cave nectar bats

To apply a consistent terminology, if a plant species was detected in (i) $\geq 8$ of the 10 weekly samples, it was considered 'frequently' detected, (ii) $>3$ but $<8$ of the 10 weekly samples, it was considered 'moderately' detected and (iii) $\leq 3$ of the weekly samples, it was considered 'infrequently' detected.

## RESULTS

A total of 47 OTU (~320 bp) were detected using *ITS2* primers and 13 OTU (~200 bp) were detected using *rbcL* primers. *RbcL* OTU which were assigned with a genus and/or family

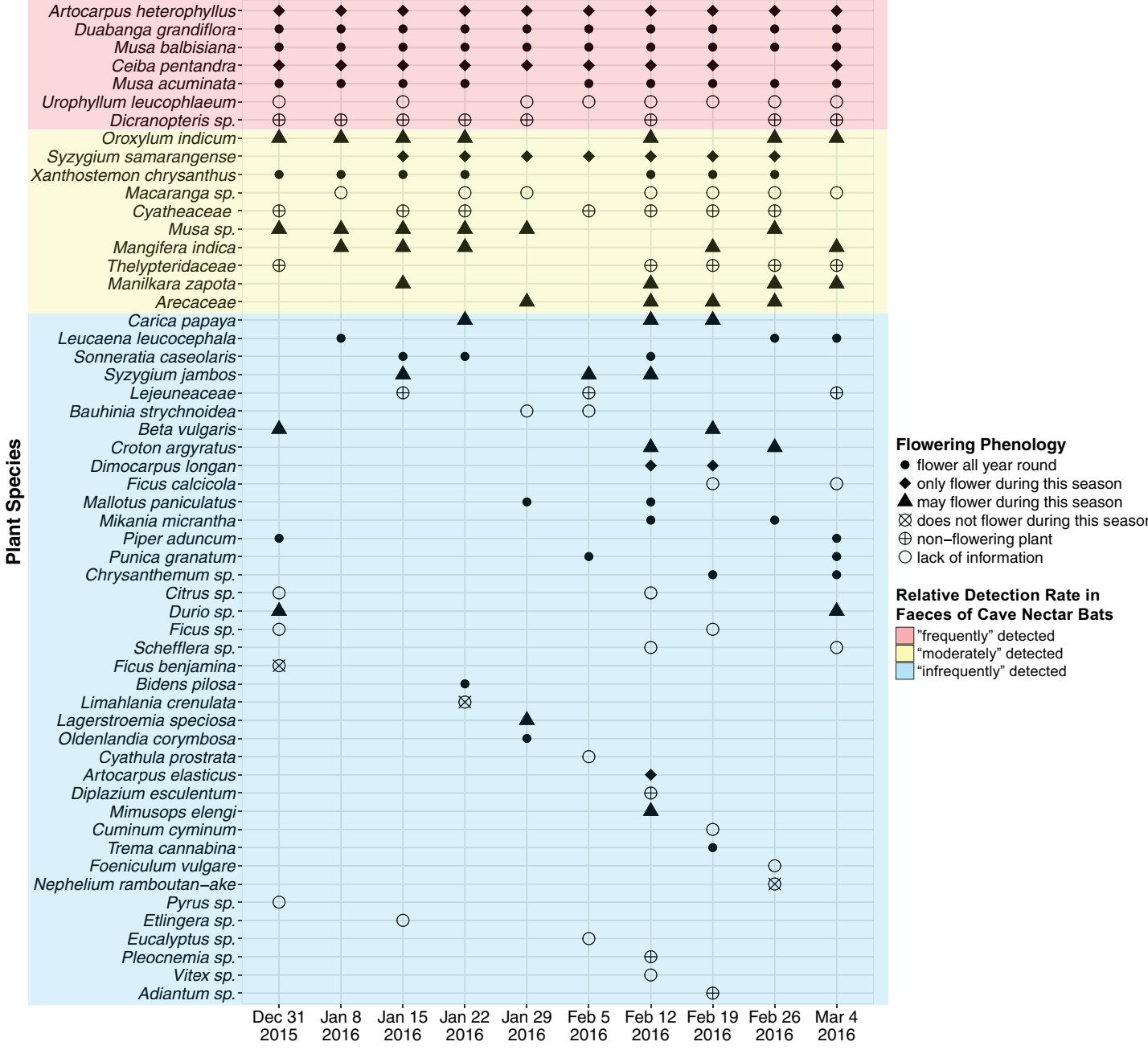

**Figure 2 Plant species detected from faecal samples of *E. spelaea* using DNA metabarcoding for 10 weeks (31st of December 2015 to 4th of March 2016).** Order of *y*-axis is based on (i) number of detection, (ii) taxonomic rank (i.e., species, genus and family), (iii) alphabetical order and (iv) date of detection.

name that was also assigned to an *ITS2* OTU were discarded as likely duplicates. This resulted in 55 OTU (*ITS2* = 47, *rbcL* = 8), of which 37 OTU were assigned a species name (*ITS2* = 36, *rbcL* = 1), 14 were assigned a genus name (*ITS2* = 11, *rbcL* = 3) and the remaining four were assigned a family name (*rbcL* = 4) (Fig. 2; Supplemental Information 2). An average of 18 OTU were detected each week (SD = 5.103, min = 12, max = 30).
**Table 1 Estimated plant richness in the faecal samples of *E. spelaea* of which the number of observed species is 55, the number of faecal sample is 10 and the total number of incidences is 185.**

| Species richness model | Estimate | Standard error | Lower limit of 95% confidence interval | Upper limit of 95% confidence interval |
|---|---|---|---|---|
| Homogenous model[a] | 60.324 | 2.996 | 56.904 | 69.882 |
| Chao2[b] | 66.604 | 7.040 | 58.873 | 89.766 |
| Chao2-bc[c] | 65.260 | 6.342 | 58.365 | 86.286 |
| iChao2[d] | 68.961 | 4.356 | 62.682 | 80.372 |

Notes:
[a] This model assumes that all species have same incidence of detection probabilities.
[b] This approach uses the frequencies of uniques and duplicates to estimate the number of undetected species.
[c] A bias-corrected form for the Chao2 estimator.
[d] Improved Chao2 estimator.

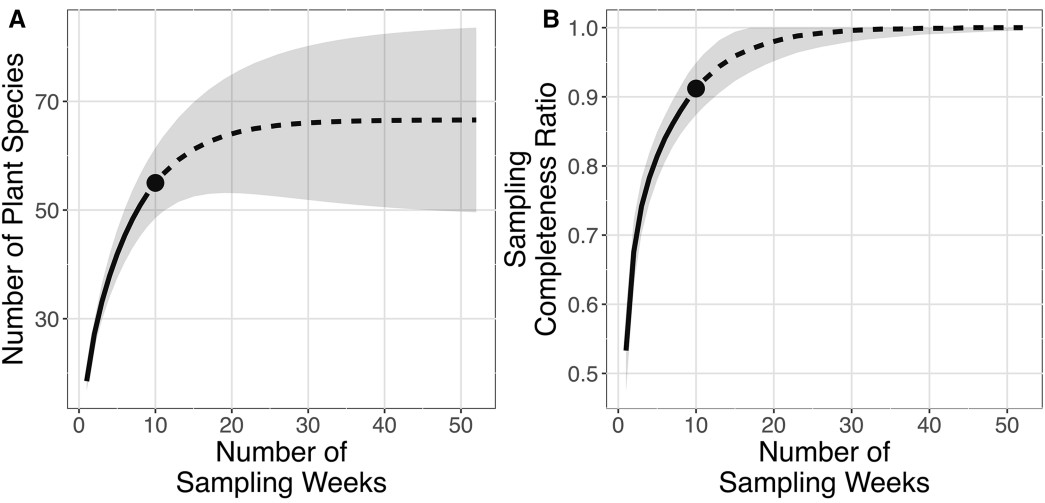

**Figure 3 Rarefaction and extrapolation sampling curves for this study (from 31st of December 2015 to 4th of March 2016) showing estimated species richness using Chao2 sampling curves are extrapolated to one year (52 weeks) with 95% confidence interval, number of replications = 100 and sampling completeness ratio = 0.912.** (A) Sample-sized-based rarefaction and extrapolation curve. (B) Sample completeness-based rarefaction and extrapolation curve.

The plant species richness in faecal samples of *E. spelaea* estimated by different models were within the range of 65.260–68.961 (Table 1). The sampling completeness ratio was estimated to be 0.912 (Fig. 3). Both asymptotic (species richness estimation) and non-asymptotic analyses (rarefaction/extrapolation) suggested that a longer sampling period and larger faecal sample size would detect more plant species in the diet of *E. spelaea* (Fig. 3).

Of the 55 plant species, 24 were native (*ITS2* OTU = 23; *rbcL* OTU = 1) while 16 were introduced to Peninsular Malaysia (*ITS2* OTU = 16) (Table 2). The status of the remaining 15 plant species is unknown (*ITS2* OTU = 8; *rbcL* OTU = 7) as we could not assign them to a species name. We detected 49 plant species which have not been reported by previous dietary studies of *E. spelaea* conducted during the same sampling months (i.e., December to March) (Table 2).

**Table 2 Checklist of plants consumed by *Eonycteris spelaea* between December and March.**

| Family | Species | Status | Type of detection[a] | Month(s) of detection[b] | References |
|---|---|---|---|---|---|
| Amaranthaceae | *Beta vulgaris* | Introduced | DNA | Dec, Feb | 4 |
| | *Cyathula prostrata* | Native | DNA | Feb | 4 |
| Anacardiaceae | *Mangifera indica* | Introduced | DNA | Jan–Mar | 4 |
| Apiaceae | *Cuminum cyminum* | Introduced | DNA | Feb | 4 |
| | *Foeniculum vulgare* | Introduced | DNA | Feb | 4 |
| Araliaceae | *Schefflera* (Unidentified) | | DNA | Feb–Mar | 4 |
| Arecaceaea | *Cocos nucifera* | Native | P | Dec–Mar | 1, 2 |
| | *Arenga* (Unidentified) | | P | Jan–Mar | 1, 2 |
| | (Unidentified) | | DNA | Jan–Feb | 4 |
| Asteraceae | *Bidens pilosa* | Native | DNA | Jan | 4 |
| Anacardiaceae | *Chrysanthemum* (Unidentified) | Introduced | DNA | Feb–Mar | 4 |
| | *Mikania micrantha* | Introduced | DNA | Feb | 4 |
| Bignoniaceae | *Oroxylum indicum* | Native | P, DNA | Dec–Mar | 1, 2, 3, 4 |
| Cannabaceae | *Trema cannabina* | Native | DNA | Feb | 4 |
| Caricaceae | *Carica papaya* | Introduced | DNA | Jan–Feb | 4 |
| Compositae | (Unidentified) | | P | Dec | 1 |
| Euphorbiaceae | *Croton argyratus* | Native | DNA | Feb | 4 |
| | *Macaranga* (Unidentified) | | DNA | Jan–Mar | 4 |
| | *Mallotus paniculatus* | Native | DNA | Jan–Feb | 4 |
| Fabaceae | *Bauhinia strychnoidea* | Native | DNA | Jan–Feb | 4 |
| | *Leucaena leucocephala* | Introduced | DNA | Jan–Mar | 4 |
| Gentianaceae | *Limahlania crenulata* | Native | DNA | Jan | 4 |
| Lamiaceae | *Vitex* (Unidentified) | | DNA | Feb | 4 |
| Leguminosae | *Parkia* spp. | | P | Dec–Mar | 1, 2, 3 |
| Lythraceae | *Duabanga grandiflora* | Native | Fl, P, DNA | Dec–Mar | 1, 4 |
| | *Lagerstroemia speciosa* | Native | DNA | Jan | 4 |
| | *Punica granatum* | Introduced | DNA | Feb–Mar | 4 |
| | *Sonneratia alba* | Native | Fl, P | Dec–Feb | 1 |
| | *Sonneratia caseolaris* | Native | Fl, P, DNA | Dec–Feb | 1, 4 |
| | *Sonneratia* (Unidentified) | | P | Dec–Mar | 2, 3 |
| Malvaceae | *Bombax anceps* | Native | Fl, P | Dec–Feb | 1, 3 |
| | *Bombax* (Unidentified) | | P | Feb | 2 |
| | *Ceiba pentandra* | Introduced | Fl, P, DNA | Dec–Mar | 1, 2, 3, 4 |
| | *Durio* spp. | Native | P, DNA | Dec–Mar | 1, 2, 3, 4 |
| Moraceae | *Artocarpus elasticus* | Native | DNA | Feb | 4 |
| | *Artocarpus heterophyllus* | Introduced | DNA | Dec–Mar | 4 |
| | *Artocarpus* (Unidentified) | | P | Jan–Mar | 1 |
| | *Ficus benjamina* | Native | DNA | Dec | 4 |
| | *Ficus calcicola* | Native | DNA | Feb–Mar | 4 |
| | *Ficus* (Unidentified) | | DNA | Dec, Feb | 4 |

| Family | Species | Status | Type of detection[a] | Month(s) of detection[b] | References |
|---|---|---|---|---|---|
| Musaceae | *Musa acuminata* (previously reported as *malaccensis* and *truncata*) | Native | Fl, DNA | Dec–Mar | 1, 4 |
| | *Musa balbisiana* | Native | DNA | Dec–Mar | 4 |
| | *Musa* (Unidentified) | | Fl, P, DNA | Dec–Mar | 1, 2, 3, 4 |
| Myrtaceae | *Syzygium jambos* | Native | DNA | Jan–Feb | 4 |
| | *Syzygium malaccensis* (previously reported as *Eugenia malaccensis*) | Native | Fl | Dec–Feb | 1 |
| | *Syzygium samarangense* | Native | DNA | Jan–Feb | 4 |
| | *Syzygium* (Unidentified) | | P | Dec–Mar | 1, 2 |
| | *Xanthostemon chrysanthus* | Introduced | DNA | Jan–Feb | 4 |
| | *Eucalyptus* (Unidentified) | | P, DNA | Feb | 3, 4 |
| Piperaceae | *Piper aduncum* | Introduced | DNA | Dec, Mar | 4 |
| Rosaceae | *Pyrus* (Unidentified) | | DNA | Dec | 4 |
| Rubiaceae | *Oldenlandia corymbosa* | Introduced | DNA | Jan | 4 |
| | *Urophyllum leucophlaeum* | Native | DNA | Dec–Mar | 4 |
| Rutaceae | *Citrus* (Unidentified) | | DNA | Dec, Feb | 4 |
| Sapindaceae | *Dimocarpus longan* | Native | DNA | Feb | 4 |
| | *Nephelium ramboutan-ake* | Native | DNA | Feb | 4 |
| Sapotaceae | *Manilkara zapota* | Introduced | DNA | Jan–Mar | 4 |
| | *Mimusops elengi* | Native | DNA | Feb | 4 |
| | (Unidentified) | | P | Feb–Mar | 1 |
| Zingiberaceae | *Etlingera* (Unidentified) | | DNA | Jan | 4 |
| Athyriaceae | *Diplazium esculentum* | Native | DNA | Feb | 4 |
| Pteridaceae | *Adiantum* (Unidentified) | | DNA | Feb | 4 |
| Dryopteridaceae | *Pleocnemia* (Unidentified) | | DNA | Feb | 4 |
| Gleicheniaceae | *Dicranopteris* (Unidentified) | | DNA | Jan–Mar | 4 |
| Thelypteridaceae | (Unidentified) | | DNA | Dec, Feb–Mar | 4 |
| Cyatheaceae | (Unidentified) | | DNA | Jan–Feb | 4 |
| Lejeuneaceae | (Unidentified) | | DNA | Jan–Mar | 4 |

**Notes:**
References: 1, *Start (1974)* reported 14 plant species; 2, *Bumrungsri et al. (2013)* reported nine plant species; 3, *Thavry et al. (2017)* reported seven plant species; 4, this study detected 55 plant species.
[a] Type of detection (Fl, sighted on and/or caught near flowers; P, pollen found in faeces and/or on body; Fr, caught near fruiting trees; DNA, DNA metabarcoding).
[b] Month of the year (Jan, January; Feb, February; Mar, March; Dec, December).

Two native plant species, *Duabanga grandiflora* and *Musa balbisiana*, and an introduced species, *Artocarpus heterophyllus*, were detected from all ten faecal samples (i.e., every week) and were flowering during the sampling period (Fig. 2; Table 2). The native *Musa acuminata* and the introduced *Ceiba pentandra* were detected in nine faecal samples, and were flowering during the sampling period. The native *Urophyllum leucophlaeum* and a fern, *Dicranopteris* sp., were detected in eight faecal samples.

## DISCUSSION

By using DNA metabarcoding to identify the plant species present in faeces of *E. spelaea* collected over 10 weeks, we detected 55 plant species, many of which had not been reported in previous studies of the diet of *E. spelaea* (including studies conducted during the same time of year; Table 2). In this study, most of the detected plants could be assigned to a species name. For example, the two OTU belonging to the economically important genus *Artocarpus* could be identified as *Artocarpus elasticus* and *A. heterophyllus*, whereas *Start & Marshall (1976)* could only identify pollen grains to the genus *Artocarpus* but could not assign to a species name. In addition, the failure of previous studies (which examined the morphology of pollen grains) to detect pollen grains of species recorded in this study may be due to degradation of the pollen grains in the bats' gastrointestinal tract (*Herrera & Martínez Del Río, 1998*). Therefore, it is difficult to conclude if the detection of these species in this study is due to the changing landscape or a result of the better detection capability of DNA metabarcoding.

In contrast, this study failed to detect several plant species that were previously recorded in the diet of *E. spelaea* (Table 2; Supplemental Information 4). This may be due to the plant DNA barcoding primers used in this study which could be biased towards the detection of particular plant families (*García-Robledo et al., 2013*; *Prosser & Hebert, 2017*). Furthermore, using BLAST (against NCBI GenBank) for OTU identification is limited to plant species which have already been sequenced and submitted to the database (*Bell et al., 2016*; *Bell, Loeffler & Brosi, 2017*). Consequently, this study may have failed to detect some of the previously reported diet species that are not currently in NCBI GenBank (e.g., *Bombax anceps* and *Syzygium malaccensis*).

The short sampling period of this study (31 December 2015 to 4 March 2016) may also account for the failure to detect certain plant species. Although the relatively high sampling completeness ratio and estimated plant species richness support the adequacy of the sampling effort for this study, both estimates only apply for the particular sampling period (when only certain plant species were flowering). As floral community changes over time (*Boulter, Kitching & Howlett, 2006*; *Delaney, Jokela & Debinski, 2015*), especially in Peninsular Malaysia where many species flower at irregular intervals (*Appanah, 1993*; *Chen et al., 2018*), a longer sampling period and larger faecal sample size will likely reveal more plant species in the diet of *E. spelaea*.

The native plant species, *D. grandiflora* and *Musa* spp. were frequently detected during our study and, considering these species flower year-round, likely represent a stable food resource for cave nectar bats throughout the year. Two other native plant species: *U. leucophlaeum*, which has been recorded in hill and montane forests in Peninsular Malaysia (*Wong, 1989*), and *Bauhinia strychnoidea*, a calciphile plant which has been recorded from Batu Caves (*Ridley, 1922–1925*), were frequently and infrequently detected in the bat faeces. Little is known about the flowering phenology and pollination ecology of these plants. The infrequent detection of a mangrove plant, *Sonneratia caseolaris*, which flowers year-round, suggests that the species is not an important food resource for these particular cave nectar bats, yet indicates that some bats likely travelled

~40 km from Batu Caves to the nearest mangrove forest at Kuala Selangor. This is congruent with the finding of *Start (1974)* who observed that individuals of *E. spelaea* roosting at Batu Caves travelled 38 km to Rantau Panjang to feed on *Sonneratia alba*. Interestingly, *Acharya et al. (2015b)* estimated the foraging range for *E. spelaea* in southern Thailand to be 17.9 km only, though this could be due to the fact that the cave roosts in that particular study were located in agricultural areas where the cultivated fruit orchards nearby provided an easy source of food. In contrast, *E. spelaea* in Batu Caves appears to travel long distances from the roost to different habitats (i.e., mangrove, limestone and montane forests) where it feeds and consequently may promote genetic diversity among plant populations by dispersing pollen (see review by *Fleming, Geiselman & Kress, 2009*). Radio-tracking the cave nectar bats at Batu Caves remains a highly desirable approach to determine their foraging distances, and assess whether the long-distance travelling behaviour (i) is sex-specific where female tend to forage further while male tend to forage closer to roost as observed in *E. spelaea* in Thailand (*Bumrungsri et al., 2013*) and *Pteropus rufus* in Madagascar (*Oleksy, Racey & Jones, 2015*) and (ii) whether it is a strategy to reduce extreme competition for food which may be a consequence of the gregarious roosting behaviour of *E. spelaea* as recognised for *Leptonycteris curasoae* in Mexico (*Horner, Fleming & Sahey, 1998*). Together these findings support the view that *E. spelaea* remain a crucial pollinator of native plants in highly disturbed habitats.

We detected many plant species which were introduced to Peninsular Malaysia and have since naturalised in the region including *A. heterophyllus* and *C. pentandra* which are commonly planted in human settlements for fruits (*Corner, 1997*) and were flowering during the sampling period. The high detection rate of these introduced plants in the bat faeces suggests that these plants may be important food resource for the cave nectar bats in human-dominated habitats. On the other hand, the moderate and infrequent detection rate of other introduced plant species which are often planted for urban beautification and shade (e.g., *Chrysanthemum* sp., *Leucaena leucocephala* and *Xanthostemon chrysanthus*) suggests that these plants may be supplement food resources for the bats (*Corlett, 2005*; *Nakamoto, Kinjo & Izawa, 2007*). However, consumption and potential pollination of these introduced plants by cave nectar bats may have an adverse impact on the reproductive success of native plants (*Morales & Traveset, 2009*) and on other dependant urban wildlife (*Corlett, 2005*; *Grimm et al., 2008*). Therefore, the status of a plant species should be considered carefully prior to gardening and landscaping activities. Planting native plants instead of introduced plants could help to promote the consumption and hence pollination of native plants by the cave nectar bats, which consequently could maintain healthy ecosystems in urban areas.

Many of the plant species detected in this study are grown as commercial food crops including jackfruit (*A. heterophyllus*), banana (*Musa* spp.), water apple (*Syzygium samarangense*), mango (*Mangifera indica*) and papaya (*Carica papaya*); most of these plants were likely to be flowering during the sampling period. One of the commercial food crops which was frequently detected and flowers seasonally is jackfruit; a fruit with an estimated production value of RM 55 million for year 2011 (*Abd-Aziz, Abd-Rahman & Razali, 2016*). The previous study in Peninsular Malaysia also reported pollen grains of genus *Artocarpus* in

faeces of *E. spelaea* (*Start & Marshall, 1976*). Altogether it is likely that *E. spelaea* play an important role in pollination of this economically important plant species.

Plant species which are pollinated and/or dispersed by wind and/or insects were detected in the bat faeces including ferns (e.g., *Adiantum* sp. and *Pleocnemia* sp.), weeds (e.g., *Bidens pilosa, Cuminum cyminum, Cyathula prostrata, Oldenlandia corymbosa*), figs (*Ficus* spp.) and *A. elasticus* (*Corner, 1997*; *Boo, Chew & Yong, 2014*). The infrequent detection of these plant species suggest they form a relatively minor part of the cave nectar bat's diet or were unintentionally consumed. It could also be likely that the spores and pollen grains of these plant species may have adhered to the fur of the cave nectar bats when they were foraging (*Corbet, Beament & Eisikowitch, 1982*) and consequently were ingested when they groomed themselves later (*Fleming, Geiselman & Kress, 2009*). Another potential explanation (though unlikely given our protocol) is that the spores and pollen grains of these plant species may have been unintentionally collected when sampling the bat faeces directly from the cave floor.

One limitation of DNA metabarcoding is the inability to identify which part of the plant is being consumed by the bats. Previous studies have observed remains of fruits and leaves in faeces and under the day roosts of *E. spelaea*, and consequently suggested that fruits and leaves may form a part of the cave nectar bat's diet (*Start & Marshall, 1976*; *Bumrungsri et al., 2013*). Similarly, we detected ferns and figs (which were either not flowering during the sampling period or have unknown flowering phenology) in the faeces of *E. spealea* but could not determine whether the bats were feeding on the fronds and fruits or ingesting the spores and pollen grains inadvertently. It is possible that *E. spelaea* chew the fronds and fruits, ingest the juice (and possibly fragments of the fronds and fruits) and spit out the fibres later; a feeding behaviour which is common in pteropodid bats including *Cynopterus brachyotis* (*Phua & Corlett, 1989*; *Tan, Zubaid & Kunz, 1998*) and *Pteropus* spp. (*Nakamoto, Kinjo & Izawa, 2007*; *Scanlon et al., 2014*; *Win & Mya, 2015*; *Aziz et al., 2017a*). The ability of pteropodid bats to eat fronds and disperse the spores of the bird-nest fern (*Asplenium setoi*) has also been demonstrated in a feeding experiment with *P. pselaphon*, an endemic to islands in Japan (*Sugita et al., 2013*). Whether *E. spelaea* is specialised nectarivore or feeds opportunistically on other parts of plants remains to be determined. Observations of *E. spelaea*'s feeding behaviour, possibly using camera traps as demonstrated in a study of the locally endangered *P. hypomelanus* (*Aziz et al., 2017b*), is a promising further avenue of research to determine (i) which part of the plants are being consumed by the bats and (ii) the interactions between the bats and plants (e.g., bats dispersing spores and seeds). Nevertheless, the use DNA metabarcoding in this study has provided important baseline data for future research into the diet of tropical nectarivorous bats.

## ACKNOWLEDGEMENTS

We thank the Cave Management Group (http://www.darkcavemalaysia.com/) for allowing us to collect samples at Dark Cave Conservation Site. We also thank Tan Kai Ren for providing assistance during fieldwork, Sugumaran Manickam and Yong Kien Thai from the Herbarium, University of Malaya (http://rimba.um.edu.my/) for their advice on

plant identification and Sheema Aziz from Rimba (https://rimbaresearch.org) for providing helpful literature and advice on data analyses. We are grateful to Ana Rainho, Alyssa Stewart, Maria Pereira and an anonymous reviewer for their comments on earlier version of this manuscript. This project was presented at the XIX International Botanical Congress in July 2017 and 22nd Biological Sciences Graduate Congress in December 2017.

### Funding

This project was supported by grants from the University of Malaya (PG060-2016A), the National Geographic Society (Asia59-16), and the Malaysian Nature Society (Young Environmental Research Grant 2016-12) awarded to Voon-Ching Lim. The funders had no role in study design, data collection and analysis, decision to publish, or preparation of the manuscript.

### Grant Disclosures

The following grant information was disclosed by the authors:
University of Malaya: PG060-2016A.
National Geographic Society: Asia59-16.
Malaysian Nature Society: Young Environmental Research Grant 2016-12.

### Competing Interests

The authors declare that they have no competing interests.

### Author Contributions

- Voon-Ching Lim conceived and designed the experiments, performed the experiments, analyzed the data, prepared figures and/or tables, authored or reviewed drafts of the paper, approved the final draft.
- Rosli Ramli prepared paperwork regarding permit and ethics approval, and reviewed drafts of the paper.
- Subha Bhassu contributed reagents/materials/analysis tools.
- John-James Wilson contributed reagents/materials/analysis tools, authored or reviewed drafts of the paper.

### Animal Ethics

The following information was supplied relating to ethical approvals (i.e., approving body and any reference numbers):

Faecal collection was conducted using a protocol approved by the Institutional Animal Care and Use Committee, University of Malaya (Ref: ISB/10/06/2016/LVC (R)).

### Field Study Permissions

The following information was supplied relating to field study approvals (i.e., approving body and any reference numbers):

Faecal collection was conducted with authorization from the Department of Wildlife and National Parks, Peninsular Malaysia (Ref: JPHL&TN(IP)100-34/1.24 Jld. 4(34)).

## DNA Deposition

The following information was supplied regarding the deposition of DNA sequences:

Raw sequence data related to this study were deposited in Sequence Read Archive (SRA) at Genbank, NCBI under accessions SAMN07956186 to SAMN07956205.

## Data Availability

The R script for estimating species richness and sampling completeness ratio has been provided as Supplemental Dataset Files.

## Supplemental Information

Supplemental information for this article can be found online at http://dx.doi.org/10.7717/peerj.4572#supplemental-information.

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
