# Peer review of "Pollination implications of the diverse diet of tropical nectar-feeding bats roosting in an urban cave"

_PeerJ, doi:10.7717/peerj.4572_

## Round 0.1 · original submission · Major Revisions

Details are required in the description of the methodology and the choice of the markers. There are questions about the discussion of the results, regarding the flower availability at that time of the year and its impact on the results, as well as some editing work to be done. Please check details below, given in the reviewers' comments.

·

Basic reporting

- The manuscript is well written, logical and easy to comprehend.

- The context and the literature references are sufficient

- The structure and the number of tables as figures seem appropriate. I would suggest nevertheless to move Table 1 to the supplementary materials, as it is not fundamental for understanding the results.

- Raw data is shared. Please consider changing the name of the file in Supplementary Info so it will be easier to understand its contents.

- Please consider being more detailed about the aims of the paper in the abstract.

Experimental design

- The research questions focus on two very different aspects - one methodological and one ecological. The methodology cannot be consider so much a novelty, so please consider giving more focus on the issue of bat pollination of exotic (invasive?) and commercial plant species. Seed dispersal of invasive plants by native animals is a timely a very relevant issue and you could exploit it better using your data.

- The ms. would benefit from information on food availability for this species. This could be inferred e.g. from the flowering season of each species. Notice that the importance of plant species based only on its occurrence during the sampling period is highly deceiving. A plant species can be preferred by this species, but if it is not easily available (because it started blooming only at the end of the sampling period) or it is too far away it will show as unimportant.

- Please consider including some information on how the different Chao 2 estimators presented differ and how those differences may be of relevance to the results.

Validity of the findings

- The authors state that the methodology employed facilitated the detection of plants that were not previously reported. It would be relevant to discuss also why previous studies recorded several species that were not detected with this novel method. Again, discussing a bit further the issue of flower availability (see above) may be fundamental to fully clarify some of the lines discussed.
Also relevant to mention if all the previous studies were performed during the same season, otherwise any comparison between studies may be less informative. Please consider including this aspect also in the Discussion.

Additional comments

'no other comments'

·

Basic reporting

This submission is a well-written manuscript using new techniques to bring new information to a long-studied topic: pollinator diet breadth. This study focuses on the diet of a nectarivorous bat species common to SE Asia, E. spelaea. Unlike previous studies, which have relied on visual identification of pollen samples, this study identifies plant resources via DNA from fecal samples. They also use their findings to examine the extent to which this common bat pollinator uses exotic versus native plant species, given that their study area was primarily surrounded by an urban landscape (with many introduced/cultivated plant species). I think this manuscript will provide useful information to the scientific community primarily for the new methodology (that is, relatively new in the field of pollinator foraging/diets), and will also be of interest to more specialized audiences (bat or pollination or urban ecologists). That being said, I also think this manuscript can be strengthened by addressing a number of issues.

Experimental design

The data collection/sampling is a bit vague, please provide more detail.
Line 117: 100 mL total? Over how many days were samples collected?
Line 126: how many samples per week?

[Note: I have limited knowledge of DNA metabarcoding, and therefore cannot assess the authors' use/description of these methods.]

Validity of the findings

My first main concern is the authors’ discussion of which plant species are “important” to the diet of E. spelaea. First of all, it can be misleading to use presence/absence data to estimate importance. A plant species may be detected every week, but it might only comprise a small percentage of the total volume consumed. Alternatively, a plant species may only be detected 2 or 3 weeks, but it might have comprised the large majority of the bat’s diet during that time. Secondly, the authors only collected samples for essentially two months (January and February). So it’s important to state that these were the most commonly encountered species during the study period, but resource use will likely differ in other months, given that available floral resources fluctuate over time (ie, different plant species flower during different times of year).

My second main issue is the discussion of their species richness results. Their analyses estimated 65-69 plant species. This is the predicted number of plant species consumed during the study period. Or you can also think of it as the predicted number of plant species consumed if the floral community did not change over time. But floral communities do change across months. If the authors were to collect samples throughout the entire year, they would likely encounter the DNA of several more plant species, and the estimated species richness would likely be greater than 65-69. This is because a large percentage of plants (even in the tropics) do not flower year round. So the calculations are not incorporating all of the species that bloom between March and December. Likewise, the reported 0.912 sampling completeness ratio is for the study period, not for the entire year.

My third main issue is more of a request, which is that the authors at least mention the possibility that not all species detected in the fecal samples are intentionally consumed. Especially the ferns, and the plant species detected only once or twice. An alternative explanation is that the bats brush against ferns (or other random plants) during foraging and their fur picks up spores/pollen, which they then ingest during grooming. Again, I wish metabarcoding could provide quantitative data, because it would be really interesting to see the percentage of each plant species in the diet.

Additional comments

One other suggestion about Lines 287-288: One idea is to find info about fruiting/flowering times of each plant species. Then you could hypothesize whether bats were eating floral resources (nectar/pollen) or fruits or leaves (if the species wasn’t flowering or fruiting during the study period).

Reviewer 3 ·

Basic reporting

See General Comments.

Experimental design

See General Comments.

Validity of the findings

See General Comments.

Additional comments

The manuscript by Lim and colleagues employs DNA metabarcoding to identify the diet of a primarily nectar-feeding species of bat roosting in urban caves in Peninsular Malaysia. The authors were able to identify 55 species of plants, from scat samples collected on the cave floor where the bats were roosting. They make the assumption that all of these plant species are part of the diet of the bat. The plant species are a mix of native and exotic species, many of the latter with economic importance. The study and the manuscript are well done and should be published in PeerJ. However, the following points should be addressed before the paper is ready for publication.

1) Metabarcoding investigations have not been restricted just to bat diets and the authors should site some additional broad and high-profile papers on using DNA barcodes to elucidate animal diets (e.g., African herbivores; neotropical insect herbivores).

2) Why were only two of the four most common plant DNA barcode markers used in this study? Both matK and trnH-psbA could greatly help in providing identifications to the species level. The authors need to justify their use of only rbcL and ITS2.

3) It is not clear why the authors assume that the DNA barcodes were recovered from digested nectar? Does nectar even contain DNA? Or is it more likely that the DNA is being recovered from pollen, fruits or other plant parts consumed and excreted by the bats?

4) The cave floor is hardly a “sterile” environment, even if it is swept clean on a daily basis. Could there be a source of contamination in the samples collected in the cave? Some of the plant species reported hardly seem like bat diet items (e.g., Bidens?).

5) A more appropriate title for the paper might be along the lines of “Evidence for pollinators from the diverse diet of tropical nectar-feeding bats inhabiting an urban cave”

6) The citations in the text need to be checked to the references (e.g., Brandon-Mong et al, 2015 absent in references).

7) BLASTing to GenBank will only bring up species with previously sequenced barcode data, which may not be the case for many tropical native species in Peninsular Malaysia. The authors may therefore be underestimating the native plant species in the bat diets.

8) There is much redundancy in the text. The authors and editors should careful edit out, especially in discussion section. For this reason the Conclusion section could be omitted as it provides nothing new.

If the authors can address these points and revise the text accordingly, the paper should be acceptable for publication in PeerJ.

---

## Round 0.2 · Minor Revisions

Please carefully read the remaining suggestions made by the reviewers before resubmitting the manuscript.

·

Basic reporting

no comment

Experimental design

no comment

Validity of the findings

no comment

Additional comments

The authors have responded to all reviewer comments, and I think the manuscript is much improved.

One last minor comment: Does Ceiba pentandra really flower year-round in Malaysia? We only observed it flowering from Nov-Feb in southern Thailand (Stewart & Dudash BIOTROPICA 50(1): 98–105 2018), and other studies have also reported it as mass-flowering.
- Flowers in August in Brazil. Gribel et al. Journal of Tropical Ecology (1999) 15:247–263
- Flowers in June-Aug in Samoa. Elmqvist et al. Biotropica (1992) 24:15-23
- “individuals that flower are highly synchronized and do so within a relatively short period of six weeks (Frankie et al., 1974; Gribel et al., 1999; Lobo et al., 2003). Along the Pacific coast of Mexico and Central America flowering occurs at the beginning of the dry season in January and February (Lobo et al., 2003).” – Lobo et al. Am. J. Bot. February 2005 vol. 92 no. 2 370-376

It might be a good idea to double-check the flowering phenology info.

·

Basic reporting

In this paper the authors examine the diet of Eonycteris spelaea roosting in an urban cave in Malaysia by identifying plant material present in bat feaces.
From my understanding this is a second review of the paper, which I did not revise previously. My understanding is that the authors answered most of the concerns of the previous reviewers and, for this reason, I feel pleased with the majority of their text and findings.

Experimental design

The experimental design seems adequate. However, the authors include information on what they refer is food availability for Eonycteris spelaea, but truly is just information on the phenology of plants that potentially belong to its diet. While I see no substantial problem with this, and understand the difficulty of getting data on food availability, the authors should be more explicit about this in their methodology.

Validity of the findings

Findings are robust and rather interesting, although the methodology is not particularly novel.

Additional comments

It would be rather relevant for the authors to suggest management actions to potentially reduce the consumption of exotic species by Eonycteris spelaea, while promoting the consumption of native species.

---

## Round 0.3 · accepted · Accept

The manuscript has been corrected as suggested by the reviewers

#